# Protruding organic surfaces triggered by in-plane electric fields

Danqing Liu[1,2], Nicholas B. Tito[3,4] & Dirk J. Broer [4,5]

Coatings with a dynamic surface topography are of interest for applications in haptics, soft robotics, cell growth in biology, hydro- and air dynamics and tribology. Here we propose a design for creating oscillating surface topographies in thin liquid crystal polymer network coatings under an electric field. By applying an alternating electric field, the coating surface deforms, and pre-designed local corrugations appear. The continuous AC electric field further initiates oscillations superimposed on the formed topographies. This effect is based on microscopic free volume creation. By exciting the liquid crystal network at its resonance frequency, maximum free volume is generated and large surface topographies are formed. Molecular simulation is used to examine this behaviour in microscopic detail as a function of oscillation frequency. Surface topography formation is fast and reversible. Excess free volume is energetically unfavourable, thus the surface topographies disappear within seconds once the electric field is removed.

[1] SCNU-TUE Joint Lab of Devices Integrated Responsive Materials (DIRM), South China Normal University, No. 378, West Waihuan Road, Guangzhou Higher Education Mega Center, Guangzhou 510006, China. [2] Department of Chemical Engineering, Delft University of Technology, Van der Maasweg 9, 2629 HZ Delft, The Netherlands. [3] Department of Applied Physics, Eindhoven University of Technology, Postbus 513, 5600 MB Eindhoven, The Netherlands. [4] Institute for Complex Molecular Systems (ICMS), Eindhoven University of Technology, Den Dolech 2, 5612 AZ Eindhoven, The Netherlands. [5] Laboratory of Functional Organic Materials & Devices (SFD), Department of Chemical Engineering & Chemistry, Eindhoven University of Technology, Den Dolech 2, 5612 AZ Eindhoven, The Netherlands. Correspondence and requests for materials should be addressed to D.J.B. (email: d.broer@tue.nl)

Surfaces are the interface between man and material, and man and device. They determine the way matter feels, smooth or rough, and matter looks, glossy or matte. In this way, they provide information by addressing human senses like touch and vision. Studies suggest that humans can distinguish changes in topographical dimensions down to the nanometer level[1]. In nature, surfaces fluctuate shape and topography in response to their local environment. They promote attraction, e.g. for propagation of genes; in other cases, they repulse, to protect against predators[2, 3]. Unlike examples in nature, man-made surfaces in general are static in their geometry and topography[4, 5]. In order to create dynamic surfaces to enhance the human interaction with surfaces, a field which is also known as haptics, responsive polymers are to be considered preferably responding to fields generated by an electrical circuit.

Well-known electrically deforming polymers, such as piezo-electric films[6], electroactive polymers (EAPs)[7] and hydrogels[8], require compliant electrodes flexible enough to deform together with the deformation. This limits the choice of electrode materials, complicates the fabrication, and even prohibits the deformations when the coating is confined at a rigid substrate. Especially the latter is needed for most haptic applications, where a coating on a solid device should provide touch information without direct user contact to the electrodes. Here we propose an approach to induce protrusions in a transparent coating by a local electrical field. This design can be easily integrated at the surface of a device.

Here we report a principle to create fluid-like dynamic topographies on a solid surface to provide haptic information to human by touch or vision. By applying an alternating electric field from a pattern of underlying interdigitated electrodes, the coating surface deforms, and pre-designed local corrugations appear. Resonance effects of a frequency-tuned electric field initiates chaotic oscillation superimposed on the formed topographies. The application potential is enormous. We envisage coatings applied on a display-inspired active matrix, in which transistor controlled pixels provide complex, but controlled 3D deformation of an initially flat surface[9, 10].

## Results

**Design and principle.** The materials used are reactive liquid crystal mesogens which are already widely applied in almost every display panel in which they have a proven durability. The mesogens are polymerized by photopolymerization after being applied as 2.5 µm thin film on the substrate with transparent interdigitated electrodes, previously made by current lithographic techniques. For activation, a lateral electric field is generated by the electrodes which are now buried under the active layer, not being touched by a potential user (Fig. 1a, b). The high dielectric permittivity of the active layer focuses the electrical field to the area between the electrodes patterns, as indicated in Fig. 1b, which determines the volume of activation.

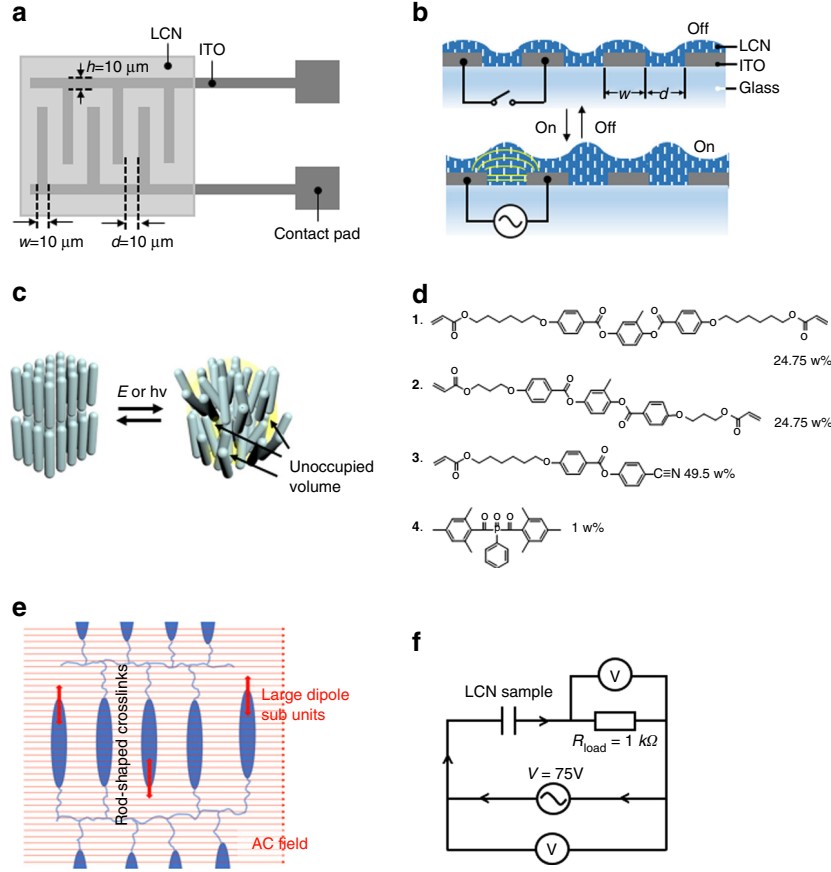

**Fig. 1** Principle of electrically-driven surface topographies. **a** Schematic representation of the interdigitated indium tin oxide (ITO) electrode array on glass and its dimensions. **b** A lateral electric field is coupled to the polymer film by the electrodes from **a**, to induce the formation of protrusions. **c** Schematic representation of the formation of molecular voids upon stress–induced reduction of molecular order in an LCN. **d** Chemical composition of the liquid crystal mixture to form the LCN by photopolymerization. **e** Schematic representation of the LCN polymer. The orientation of the LC molecules perpendicular to the AC field provides the torque that induce the oscillatory stresses. **f** The electric circuit employed drives the sample deformation and measures the current flow. The LCN coating stores energy when placed in the external electric field and functions as a capacitor. A resistor with known resistance is in series with the sample to measure the current by means of an oscilloscope. An alternating voltage source drives the sample and the actual applied voltage is measured

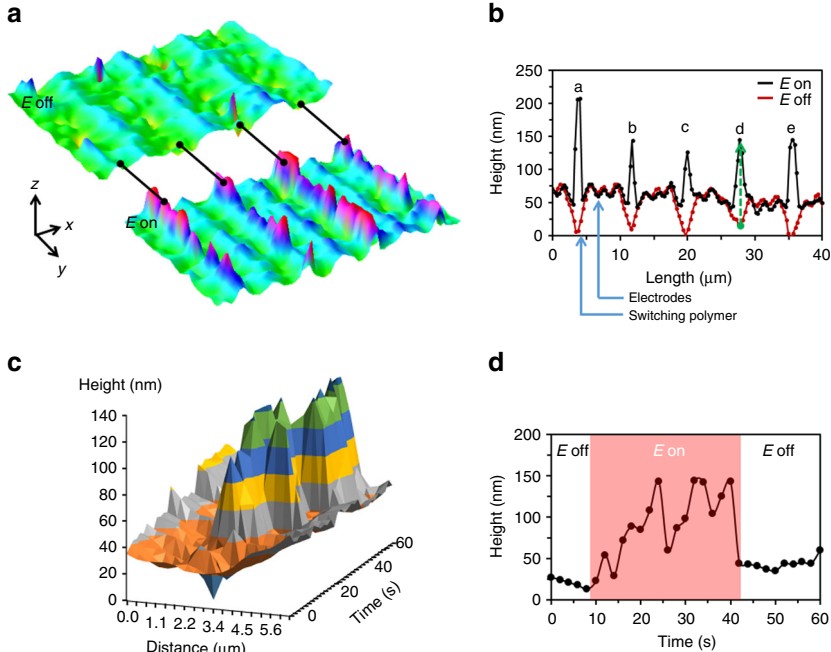

**Fig. 2** Surface deformations under an electric field. **a** 3D images of a coating on an electrode array in its initial state, and when the electric field is on. **b** The corresponding cross-sectional height profile of **a** measured in x-direction. The red curve gives the profile in the field-off state, with somewhat elevated areas over the electrodes. The black line is a snapshot measured during the field-on state showing a protrusion of around 150 nm. The sample is activated at 41 °C under an electric field strength of 7.5 V/μm at 900 kHz. **c** Time evolution of profile "d" in **b** from the initial to the activated state under a continuous AC electric field. **d** The height change over time of the selected point marked green in profile "d" of **b**

The principle of surface deformation is based on the creation of molecular voids by bringing dynamic disorder in liquid crystal polymer networks (LCNs) (Fig. 1c). Previously, we demonstrated that free volume (i.e., non-occupied volume) is created when the polymer network of well-ordered molecular mesogens is subjected to an oscillatory molecular stress at a specific frequency[11]. Azobenzene crosslinks addressed by dual wavelength light provided push/pull effects on the network reducing the order parameter slightly and creating temporarily unoccupied volume, which deforms the surface. The principle that we employ here to induce topographies is inspired by these recent findings. But rather than bringing the polymer network into oscillation by light, we use an AC electric field to exert an oscillatory stress on the LCN main chains. The mesogenic rods are thereby continuously changing their initial orientation and packing which leads to free volume increase[12]. The addressing AC frequencies are tuned to natural network frequencies initiating resonance-enhanced oscillatory dynamics which largely amplify the deformation effects.

**Materials and processing**. The LCN films are made by in-situ polymerization of reactive mesogens given in Fig. 1d[13, 14]. We have chosen a set of two liquid crystal diacrylates (**1** and **2**) to form a polymer network and a cyanogroup containing monoacrylate (**3**) that couples strongly to the electric field by its polar end group. A thin film is formed by spin coating from solution. The substrate is pretreated, providing an alignment of the molecular rods perpendicular to the substrate surface. The film is photopolymerized by UV light that excites photoinitiator (**4**) to form the LCN with a typical order parameter between 0.6 and 0.7. The choice of monomers and their ratio determine the properties both in the monomeric and polymeric state. The thin film coating as produced is in its glassy state at room temperature (RT) and has a wide glass transition temperature range starting at 60 °C. Eventually, these properties can be further optimized by ratio and chemical adjustment of the monomers.

The molecular alignment of the polar moieties is chosen to be perpendicular to the AC field lines which couples in a maximum oscillatory torque, as schematically shown in Fig. 1e. The AC field exerts force on the molecules but is not able to fully realign the molecules[15] as known for low molar mass liquid crystals used in displays. The rigidity of the polymer network prohibits this.

**Actuation and characterization**. Digital Holographic Microcopy (DHM) is used to characterize surface topographies (Supplementary Fig. 1). Prior to actuation, a small surface relief of around 60 nm is observed. This is because the thin LCN coating follows the contour of the underlying ITO patterns. During actuation, the sample is connected to an electric circuit as shown in Fig. 1f. The expansion measured between the electrodes reaches up to 6% of the initial coating thickness (Fig. 2a, b and Supplementary Movies 1–3; more surface profiles are shown in the Supplementary Fig. 2). This is calculated as the height change between the 'on' and 'off' states over the activated region (150 nm) divided by the initial coating thickness of 2.5 μm at the same location.

Remarkably, the experiments reveal an additional chaotic oscillation in the electrically expanded area far beyond the noise level of the measurement. To gain insight into the formation, oscillation, and relaxation of the surface topographies, we isolated one active area between electrodes (profile **d** in Fig. 2b) and followed its cross-section profile evolving in time (Fig. 2c). Upon actuation, not only the magnitude changes, but also the shape of the profile and the relative position in the x-direction. This oscillating effect is shown in Supplementary Movie 4. To elaborate this further, we selected a single point at an arbitrary distance of 3.07 μm in Fig. 2c and monitored the relative height change over time under the influence of the electric field. As seen in Fig. 2d, the activated protrusion shows a random vibration under the continuous AC field. Summarizing, the vibrations develop both in the x and y directions as well as in time with a frequency that is orders of magnitude lower than the frequency of the AC field. This

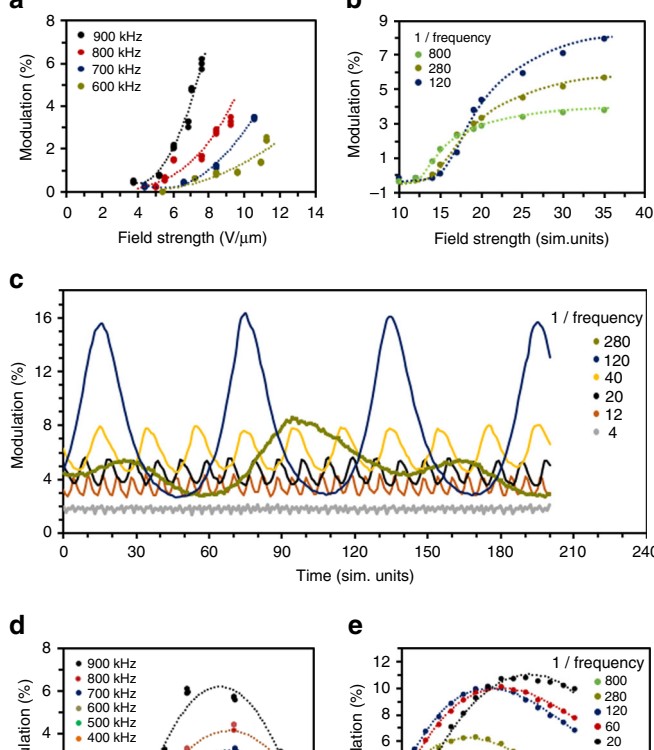

**Fig. 3** LCN deformation details. **a** Measured deformation increases with increasing field strength. The sample is at 46 °C. **b** Simulated volume modulation as a function of field strength at fixed temperature $T = 0.35$ sim. units. **c** Modeled percent volume change through simulation time in the presence of an alternating electric field. Percentage relative to the volume of the sample before the field is applied (see Supplementary Eq. 3). Results are given for different field frequencies (in sim. units), at fixed temperature ($T = 0.35$ sim. units) and field strength ($E = 30$ sim. units). **d** Influence of frequency on measured modulation at various temperatures, voltage is kept at 75 V. **e** Volume modulation in simulation as a function of temperature, at fixed electric field strength of $E = 30$ sim. units and different field frequencies (in sim. units). Units of measure in simulation are described in the Supplementary Note 6

oscillation continues until the electric field is turned off, at which point the protrusion disappears and the initial state is recovered. Figure 2b is snapshot of a deformation taken at an arbitrary time duringbthe field-on state. Because of the chaotic oscillations, which are random also with respect of their location, the deformation as shown in Fig. 2b may vary in height.

**Electrical field strength and frequency.** To understand how the microscopic changes involved in the dynamic surface topographies, we carried out molecular dynamics simulations (Supplementary Note 6)[16, 17]. A broad range of temperatures, electrical field strengths, and electric field frequencies are examined, to access the qualitative behaviour of the system at the microscale. We do not make an explicit mapping between simulation and experimental units of measure. Nevertheless, over the range of parameter space we explored in the course-grained simulations, non-trivial trends are revealed that closely mirror those seen in experiment. Our simulations suggest that the formation of dynamic surface topographies originates from free volume generation, where order parameter reduction and harmonic resonance work in concert. Without the

electric field, the mesogens of the LCN are highly ordered and compact. When an alternating electric field is applied, dielectric interaction between the field and the mesogens with large dipole moment (for dielectric properties see Supplementary Fig. 3) promotes network vibrations and molecular-scale perturbations. Subsequently, the rod-like mesogenic units deviate from their initial alignment leading to an order parameter decrease and nano-voids (molecular voids) are generated, also known as dynamic free volume[13, 18]. This is directly observed in our simulations (Supplementary Movie 5).

The simulations reveal that when the electrically-induced oscillation frequency coincides with a certain natural frequency of the LCNs between the electrodes, resonance effects occur, which significantly boost the formation of free volume, a phenomenon that was observed earlier in experiments at light driven systems[13]. The simulation do not directly give an evidence for the chaotic behaviour of the protrusions. However, to postulate an influential factor one might argue that the resonance frequency of the material depends on many factors such as temperature, modulus, density and even shape[19]. All these parameters change when the protrusions form. Consequently, the resonance conditions change upon actuation, which creates a feedback loop that switches resonance on and off. Continually repeating this cycle maintains, the oscillatory fluctuations of the surface topographies, until the electric field is stopped. We anticipate, that this is a possible origin for the chaotic oscillations we observe on top of the overall surface deformation between the electrodes.

We have examined the influence of electric field strength and frequency on the deformation, both in simulation and in experiment. The height of the surface topographies can be adjusted by the electric field strength, as seen in Fig. 3a. This is intuitively understood in terms of the larger torque (Fig. 1e) exerted on the molecules with increasing field strength. Figure 3b carries out an analogous comparison in simulation. Like in Fig. 3a, we see qualitatively that the time averaged percent volume change of the sample (at fixed temperature) increases with increasing field strength. The total volume of the material in simulation fluctuates through time around a mean value, as shown in Fig. 3c. The amplitude, period, and mean value of the fluctuations depend on the electric field frequency, strength, and system temperature. In Fig. 3c, the latter two parameters have been fixed; in that scenario, we find an "optimal" electric field frequency near 1/120 sim. units which leads to the largest sample volume fluctuations. This frequency corresponds closely to the fundamental resonant frequency of the simulated volume element itself. Increasing or decreasing the electric field frequency causes the field to be off-phase from this intrinsic material mode, leading to less optimal (and seemingly "chaotic") volume fluctuations when the conditions or material parameters change.

Both the experiment (Fig. 3d) and the simulation (Fig. 3e) indicate the existence of a threshold frequency. Lower frequencies do not trigger deformation. This further supports the notion that the topographies originate from oscillatory-free volume generation, rather than from e.g. thermal expansion (Supplementary Fig. 4). We also observe an increase of volume response with frequency (Fig. 3c, d). In the experiments, the electric field frequency does not exceed 900 kHz due to the equipment limitation. However, our molecular simulations suggest (Fig. 3c, Supplementary Fig. 8) an upper limiting frequency at which molecules can still (just barely) follow the oscillating field polarity; above this limit, the molecules begin to lag behind the field oscillation, and free volume formation decreases.

At each individual frequency, the sample is also subjected to a temperature sweep from 26 °C to 90 °C. We found an optimal temperature around 62 °C at which the largest deformation is reached (Fig. 3d). The deformation is smaller either below or

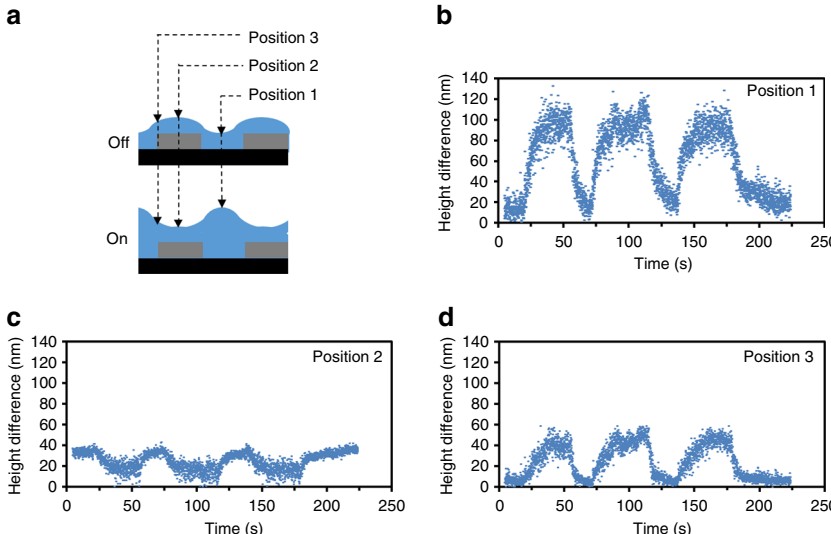

**Fig. 4** Formation and relaxation kinetics of surface topographies at various positions. **a** Schematic cross-sections representing the LCN coating at rest ('off'), and under the influence of the electric field ('on'). **b–d** Dynamics of height changes in the LCN film when the electric field is switched 'on' and 'off' for three cycles taken from a durability test of 1200 switching times. The measurement locations for each dataset are indicated in **a**. The scattering of the data correspond to the chaotic oscillating dynamics. The sample is activated at 41 °C under an alternating electric field of 7.5 V/µm at 900 kHz.

above this optimum temperature, even at high frequencies and high voltages. This maximum is also predicted by simulation (Fig. 3e). To understand this variation in volume response with temperature, we characterized the mechanical properties of the film by performing Dynamic Mechanical Thermal Analysis (DMTA) measurements (Supplementary Fig. 5). The storage modulus of the LCN film is around 2 GPa at room temperature where volume formation is restricted by the glassy state of the network. At temperatures in the rubbery state of the network, volume formation is hindered by the high network mobility, where the molecular units rapidly fill empty spaces. The glass transition is measured between 60 and 120 °C, which is close to the temperature, where we found maximum surface deformation. It is also in this region where in classical polymer physics energy dissipation is known to become maximized. For our purpose, the network mobility is just high enough to form the dynamic voids but too low to fill the empty space quickly. This phenomenon has been observed in light-driven deformations of LCNs before[20].

**Kinetic of protrusion formation and release**. In order to quantify the deformation kinetics, we measured the time-resolved electro-mechanical response at the three locations given in Fig. 4a. These positions are of specific interest with respect to the distribution of the electric field lines. The center position **1** is where the in-plane field strength is at the maximum and the polymer is subjected to the largest force. Therefore, at this location, the largest deformation is observed (Fig. 4b). Position **2** in Fig. 4a is located on top of the electrodes, where the coating is not subjected to measurable in-plane electric field; thus, the surface is expected to remain unchanged. However, the measurement in Fig. 4c shows a negative deformation. We anticipate that this is caused by the mechanical stress-strain from the expanding connected areas. We observed an intermediate deformation at the edge of the electrodes, given by position **3** in Fig. 4a. At this position, the field strength is lower, the field line is bent, and the local material suffers mechanical restrictions.

The response time of formation and relaxation of surface topographies in our LCN coating are of the order of 10 s. This is much slower than the rise and decay times of the electric field. This suggests that the deformation dynamics are dominated by

viscoelastic deformation of the polymer network, rather than by dielectric properties. Related observations in azobenzene-modified LCNs support this conclusion[13].

## Discussion

We describe a method to create dynamic surface topographies by an in-plane electrical field. Fueled by a continuous AC field, optimized for oscillating resonances between LCNs and field, the coating shows the formation of protrusions with a superimposed chaotic surface oscillation. The protrusions disappear as soon as the field is switched off. As supported by course-grain simulations, the mechanism is based on the generation of temporarily unoccupied free volume when the dielectrically anisotropic layer is subjected to an oscillating electrical field. The effect maximized when resonance conditions are found between the oscillation frequency of the field and some eigen frequency of the polymer network embedded between the electrodes.

For the electrode structure, we selected a design comparable to those used for some active matrix displays. It localizes the field and thus the deformation at a desired position. The driving voltages are not too far off of those generated by the transistors in active matrix displays, which opens a road map to surfaces than can deform controlled in time and place. It solves an outstanding problem to make coating surfaces suited for haptics[21].

The protrusions formed are of the order of hundred nanometers, just noticeable by a human finger. Among others, their heights are limited by the thickness of the LCN film, which is a few micrometers. On the longer terms, we anticipate that the formed surface structures can be enhanced further such that applications as braille displays[22] can be developed. In this paper, the effect of the surface topography formation is maximized around the glass transition temperature of the coating which is around 80 °C. For practical application, the composition needs to be optimized further, e.g., by choosing monomers with longer alkylene spacers, to reduce the glass transition temperature and thereby the temperature of use.

## Methods
**Materials**. Figure 1c shows the components for our reactive mixtures. Liquid crystal monomers **1**–**3** were obtained from Merck UK. Photoinitiator **4** was purchased from Ciba Specialty Chemicals. Typically, thin films were fabricated from a

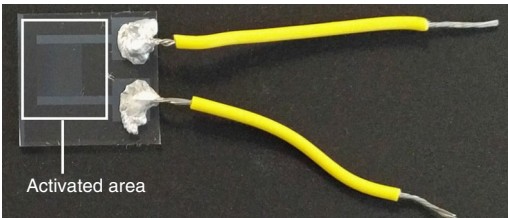

**Fig. 5** The photo of a sample. Glass substrate is patterned with IDE. LCN is coated on top of the electrodes (highlighted by the white square). Two wires are attached to the contact pad of the IDE to connect to the power source.

mixture with a ratio 1:1:2 for **1**, **2**, **3**, respectively. The constituents were mixed by dissolving in dichloromethane (DCM). Differential scanning calorimetry (DSC) results suggest that the mixture is nematic in the temperature range between 25 and 40 °C above which is isotropic.

**Device configuration**. The localized in-plane electric field is provided by an array of interdigitated ITO electrodes on a glass substrate. The LC mixture is spin coated on this substrate. After photo-polymerization a thin solid glassy polymer film is obtained with a thickness of 2.5 μm. The electric activated area has the dimension of 0.5 × 0.5 cm. Figure 5 is the photo of a sample.

**Sample preparation**. Glass substrates are cleaned by a 5 min dip in acetone under stirring, 5 min in propanol-2 under stirring, flushed with demi water followed by drying with a nitrogen flow. AL7511 (Sunever, Nissan Chemical, Japan) was used to obtain homeotropic alignment of the liquid crystal monomer mixture. It was spin coated on cleaned glass, followed by baking at 200 °C for 1 h. The film thickness of the polyimide is between 20 and 30 nm. A 20 w% concentration of monomer mixture dissolved in dichloromethane was spin coated on treated glass plates. Spin coating conditions are: speed 1500 rpm, acceleration 50 r.p.m.s-1, duration 30 s. After evaporation of the solvent, the mixture was cured by UV exposure at room temperature under nitrogen using a mercury lamp (EXPR Omnicure S2000) for 5 min. After photopolymerization, the samples were post-baked at 120 °C under nitrogen for 10 min to ensure full cure of the acrylate monomers. The final coating thickness is 2.5 μm.

**Sample characterization**. The homeotropic film was checked on orientation and eventual defects by polarized microscopy (Leica). The alternating electric field with the sinoidal wave function is provided by the function generator (AFG3252C, Tektronix). The electric signal from the function generator is amplified through an amplifier (F20A, FCL electronics). The output voltage is measured by an oscilloscope (DSOX3032 T, Keysight). The surface topographies are measured by Digital Holography Microscopy (Lyncée Tec SA.).

**Data availability**. The data are available on request from the authors.

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

## Acknowledgements

The results presented are part of research programs financed by 4TU HighTech Materials research programme 'New Horizons in designer materials' (www.4tu.nl/htm) and the European Research Commission under ERC Advanced Grant 66999 (VIBRATE), Guangdong Innovative Research Team Program (No. 2013C102), and NWO VENI grant 15135. We thank Michael Wübbenhost for the dielectric measurement. We appreciate Tristan Putzeys for his help setting up the electric circuit. We also thank Merck KGaA for providing IDE substrates. N.B.T. would like to thank Wouter Ellenbroek and Cornelis Storm for insightful discussions on this work. N.B.T. is also grateful for the compute time obtained from the NWO SURFsara Pilot Program.

## Author contributions

D.J.B.: Designed the project and wrote the manuscript. D.L.: Performed the experiments and wrote the manuscript. N.B.T.: Performed simulations and wrote the manuscript.

## Additional information

**Competing interests:** The authors declare no competing financial interests.

