## [Peer Review File · Nature Communications]

Reviewers' comments:

Reviewer #1 (Remarks to the Author):

The authors propose a new design for creating oscillating surface topographies in thin liquid crystal polymer network coatings under an electric field. Chaotic protruding has been observed depending on temperature and frequency. The results shown are well-organized, and convincing according to their previous literatures such as 2016 Nat. Commun. The concept of designing oscillating surface using the LCN by an electric field is indeed new, as an extension of their work. I recommend publication of this manuscript in Nat. Commun.

- (1) Is the surface oscillation partly based on dynamic scattering mode in the LCN?
- (2) Is the surface expansion due to an order-disorder change in molecular alignment?
- (3) How does heat generated by electric-field application affect the surface structuring?
- (4) Does the surface structural change take place below T_g at high frequencies at high voltages?
- (5) Is it possible to visualize real-time molecular alignment changes by polarized optical microscopy?
- (6) In Fig. 4, experimental conditions such as temperature, voltage, frequency, and so on should be described.
- (7) How many cycles are the LCN films durable?

Reviewer #2 (Remarks to the Author):

The authors have shown that the surface of thin liquid crystal polymer network coatings can be modulated by applying an alternating electric field. Before considering possible publication in Nature Communications, they need to provide answers on the following points.

- 1) They have shown molecular simulation results with the oscillation frequency as a parameter (not as functions of the oscillation frequency). Their numerical results shown in Figs. 3b and 3e were plotted with ambiguous 'sim. units' as the input variables instead of real physical parameters. The frequency for numerical calculation ($1/\text{frequency} = 800$ with no units) is very different from experiments (Figs. 3a and 3d, $f = 900$ kHz) without any explanation on this. Therefore, it is not easy to compare numerical results with experimental results. Please provide clear explanations on this.
- 2) They explained that maximum free volume is generated and large surface topologies are formed by exciting the liquid crystal network at its resonance frequency. However, they have not shown any results which confirm this claim. Please provide clear proofs on this.
- 3) They observed chaotic oscillation behavior. However, they did not provide any comments on the observed chaotic behavior. I recommend them to provide clear explanations on why this chaotic behavior happens.
- 4) They have shown very high spikes in the measured surface profile (Fig. 2b) when an in-plane electric field is applied to the sample. However, they did not provide any comments on the observed spikes. I recommend them to provide clear explanations on why this can happen.
- 5) As for the explanation on Fig. 3c 'However, we predict from simulation an upper limiting frequency above which the resonance conditions increasingly fail to meet (Fig. 3c, Fig. S8)' please provide more clear explanations on numerical results shown in Fig. 3c.
- 6) As for the caption of Fig. 3f 'The equivalent electric circuit employed to drive the sample,' Fig. 3f is not the equivalent circuit of the driving circuit. It is the circuit used to measure the current flow through the LCN sample.
- 7) They need to more detailed explanations on their experimental and numerical results shown as figures. They need to provide comments on whether the obtained results are the same as they expected or not. If there exist differences, they need to provide explanations on the cause of the differences.

Reviewer #3 (Remarks to the Author):

This paper represents important, pioneering work in switchable topography. The findings, while quite "early" and unrefined (in terms of technological relevance), provide a direction for work that can have very broad utility in the human-computer and even the broader human-product interface. I do have some technical questions that, when answered, will improve the paper by anticipating similar questions from Nature's readership.

1) Before reading the manuscript, coming from the wrinkling/soft matter community, I anticipated that the spatial dimensions and patterns of the switchable features would be a material property. Instead, if I understand the findings correctly, the spatial features are determined by the underlying ITO electrode pattern to which the LCN coating conforms. Assuming this is true, that aspect should be emphasized and associated limitations articulated.

2) The paper alludes to haptic interface, but the changes in topography achieved are not obviously within a regime of human tactile perception. One reference in the introduction suggests that this is true, but the matter should be addressed more rigorously. The last word of the manuscript is "Braille", suggesting that the findings will be enabling for dynamic Braille, but the difference in topography amplitude between standard Braille and this paper are about 6 or 7 orders of magnitude. As such, use of the term so loosely should be avoided and some rationale for the potential provided.

3) The explanation of the phenomena observed is rooted in a volumetric effect, particularly electrically (AC) induced free volume addition between interdigitated electrodes. This is plausible and backed up by molecular dynamics simulations included in supplementary matter. As no experimental evidence of a free-volume enhancement effect is presented, it would be prudent to offer language less strong about the explanation than that provided in the paper. As written, the readers may gain the false impression that the mechanism at work has been proven, which it has not.

4) Importantly, the effect is found to be maximized at the glass transition temperature for the LCN used, which is about 60 C. As this would not be directly useful for devices or products operating at or near room temperature, it would be helpful for the authors to indicate the feasibility of achieving molecular designs that would yield the phenomenon at or near room temperature.

5) Related to (4), the authors indicate that because the observed topography switching occurs under AC (not DC) excitation that features resonance, the effect is not one of simple thermal expansion. While plausible, it would be helpful if the authors reported some proof that the application of resonant-condition AC fields did not increase the coating temperature substantially, perhaps by thermal imaging.

If these issues can be addressed, then the paper would promise to be significant and impactful.

Answers to the reviewers' remarks:

Reviewer #1 (Remarks to the Author):

The authors propose a new design for creating oscillating surface topographies in thin liquid crystal polymer network coatings under an electric field. Chaotic protruding has been observed depending on temperature and frequency. The results shown are well-organized, and convincing according to their previous literatures such as 2016 Nat. Commun. The concept of designing oscillating surface using the LCN by an electric field is indeed new, as an extension of their work. I recommend publication of this manuscript in Nat. Commun.

1. Is the surface oscillation partly based on dynamic scattering mode in the LCN?

Answer:

DSM phenomenon has been observed widely in low molecular weight liquid crystals and we are well-known with this phenomenon as it is part of another research program in our group. However, in our LCNs we don't observe DSM effect and it is unlikely that they can exist in our coating for the following arguments:

- (1) In order to initiate electro-hydrodynamic effect, the LC mixture should be in its monomeric state/ low molecular weight state. Our coating is a densely cross-linked polymer network which has not the mobility for the electro-hydrodynamic effect as all mesogens are connected to the network.
- (2) The driving frequency is far off. Normally DSM operates at low frequency (~50 Hz), at which the present ions are excited by the low frequency alternating current (AC) and are brought into a dynamic convection-like motion. At higher frequency, e.g. at a few kHz, and also at lower voltages the widely studied dielectric re-orientation of the LC molecules occurs presently applied in displays. Our coating is driven at the frequency

(900 kHz) few orders higher than the regime of DSM. Also, realignment of our mesogens does not occur other than by a small reduction of the order parameter. As we consider DSM beyond the scope of our research we did not add a remark on this.

2. Is the surface expansion due to an order-disorder change in molecular alignment?

Answer:

Reviewer is right, although the coatings do not undergo a full order-disorder phase transition. A small decrease in order parameter from 0.6 to 0.5 is already sufficient to induce volume increase in LCNs. We have emphasized this better in the main text: on page 3 in the 2nd paragraph and on page 8 in the 2nd paragraph.

3. How does heat generated by electric-field application affect the surface structuring?

Answer:

Indeed, heat generation in the circuit is potentially an important factor. We therefore dedicated a special section to this in the Supplementary Information. We followed two approaches to estimate the heat effects, results have been given in Supplementary Section, section "Sample heating and thermal effects" and the related Figure S4. To summarize the results, we have measured the current generated in the circuit and calculated from there the estimated temperature increase to be 8 °C. Also, we have used an infrared camera to record temperature increase and measured to be 6 °C after 30 seconds after which an equilibrium was established. The measured temperature increase of 6°C contributes to a linear thermal expansion of 6.4 nm which is only 3 to 6 % of the total deformation measured (100 to 150 nm). With respect to the thickness of the initial film it is only 0.12%. Therefore, we conclude that temperature effect plays a minor role on the overall deformation.

4. Does the surface structural change take place below T_g at high frequencies at high voltages?

Answer:

We observed the surface deformation below T_g at high frequencies at (relatively) high voltage. However, the effect is smaller than around the T_g. We emphasized the role of the T_g better on page 10, 2nd paragraph.

5. Is it possible to visualize real-time molecular alignment changes by polarized optical microscopy?

Answer:

This is a good suggestion and we tried. However, we have only a small reduction in order parameter from 0.6 to 0.5. Nevertheless, under the AC field one could expect a small preferential orientation. But it appeared that there is no observable uniaxial re-alignment of the molecules, but it seems more a random reorientation in all directions. Between crossed polarizers the sample is black because of the homeotropic orientation and remains black when electrically addressed.

6. In Fig. 4, experimental conditions such as temperature, voltage, frequency, and so on should be described.

Answer:

We have added the required information in the caption of Fig.4.

7. How many cycles are the LCN films durable?

Answer:

In the experiments, we have performed six months lifetime test on the sample. We have switched the sample on/off more than 1200 times over a period of 6 months. The measurements shown in Fig. 4 are taken from these series. We included this information in the caption of Fig.4.

Reviewer #2 (Remarks to the Author):

The authors have shown that the surface of thin liquid crystal polymer network coatings can be modulated by applying an alternating electric field. Before considering possible publication in Nature Communications, they need to provide answers on the following points.

1. They have shown molecular simulation results with the oscillation frequency as a parameter (not as functions of the oscillation frequency). Their numerical results shown in Figs. 3b and 3e were plotted with ambiguous 'sim. units' as the input variables instead of real physical parameters. The frequency for numerical calculation ($1/\text{frequency} = 800$ with no units) is very different from experiments (Figs. 3a and 3d, $f = 900$ kHz) without any explanation on this. Therefore, it is not easy to compare numerical results with experimental results. Please provide clear explanations on this.

Answer:

In simulation, we have sampled a broad range of electric field frequencies, strengths, and system temperatures, in order to examine the qualitative molecular behavior of the system. The model we employ is coarse-grained, and has not yet been quantitatively mapped to experimental units. Nevertheless, we capture non-trivial trends in our simulations that agree well with those seen in experiment.

We have included an additional paragraph (Page 9, 2nd paragraph) to clarify that the simulations are not quantitatively mapped into experimental units. We have also added a sentence at the end of the caption of Figure 3 pointing the reader to the Supplementary Materials for more detail on the units of measure used in simulation.

2. They explained that maximum free volume is generated and large surface topologies are formed by exciting the liquid crystal network at its resonance frequency. However, they have not shown any results which confirm this claim. Please provide clear proofs on this.

Answer:

Theory predicts an explicit frequency at which the effect is maximized. This leads to the conclusion that the external driving frequency matches with some eigen frequency of the LCNs as the molecular dynamics predicts. Experiments show that a threshold frequency exists above which the surface starts to deform. From there the effect steeply goes up. The frequency at which the effect maximizes could not be found because of restrictions of the existing function generators. In addition, the fact that the deformed topographies start to oscillate supports this theory further. We have postulated this theory better on page 9, 1st and 2nd paragraph.

3. They observed chaotic oscillation behavior. However, they did not provide any comments on the observed chaotic behavior. I recommend them to provide clear explanations on why this chaotic behavior happens.

Answer:

The chaotic oscillation we explained by the changes of temperature, modulus, and density during the actuation step. The sample that first met the resonance conditions fails to do this as the conditions change. But as soon the deformation falls back to its initial state the resonance conditions are met again, thus forming a loop. This is explained on page 9, 1st paragraph. In addition, we cannot exclude that also other resonant modes play a role. This described in the 2nd paragraph of page 9.

4. They have shown very high spikes in the measured surface profile (Fig. 2b) when an in-plane electric field is applied to the sample. However, they did not provide any comments on the observed spikes. I recommend them to provide clear explanations on why this can happen.

Answer:

The spikes are less significant than reviewer indicates. It is somewhat exaggerated by the axes dimensions. The highest deformations are 50 nm higher than the average deformations and can be partly explained by the snapshot we take while the surface is oscillating. This is better explained in text just above Figure 2. Another explanation could be the presence of irregularities in the coating. To study this further we made reproducibility studies by testing several samples. To give an indication on the spreading in results we provided other surface profiles in the Supplementary Fig. S2.

5. As for the explanation on Fig. 3c 'However, we predict from simulation an upper limiting frequency above which the resonance conditions increasingly fail to meet (Fig. 3c, Fig. S8)' please provide more clear explanations on numerical results shown in Fig. 3c.

Answer:

In the experiment, we observe an increasing deformation with increasing frequency. Due to the experimental limits, we cannot reach higher frequency. However, we can predict now from simulation that there exists an optimal frequency, above which the deformation decrease. We changed the text in paragraph 11 to make this clearer.

6. As for the caption of Fig. 3f 'The equivalent electric circuit employed to drive the sample,' Fig. 3f is not the equivalent circuit of the driving circuit. It is the circuit used to measure the current flow through the LCN sample.

Answer:

The reviewer is right that the circuit is designed to measure the current flow. The current flow is measured at the same time as the sample is driven. Therefore, the circuit serves both measuring current flow and driving the sample. We have made this point clearer in the caption of Fig. 1f.

7. They need to more detailed explanations on their experimental and numerical results shown as figures. They need to provide comments on whether the obtained results are the same as they expected or not. If there exist differences, they need to provide explanations on the cause of the differences.

Answer:

Our simulation is to predict the trend rather than explicit mapping with the experimental results. We have now explained this better in paragraph 9.

Reviewer #3 (Remarks to the Author):

This paper represents important, pioneering work in switchable topography. The findings, while quite "early" and unrefined (in terms of technological relevance), provide a direction for work that can have very broad utility in the human-computer and even the broader human-product interface. I do have some technical questions that, when answered, will improve the paper by anticipating similar questions from Nature's readership.

1. Before reading the manuscript, coming from the wrinkling/soft matter community, I anticipated that the spatial dimensions and patterns of the switchable features would be a material property. Instead, if I understand the findings correctly, the spatial features are determined by the underlying ITO electrode pattern to which the LCN coating conforms. Assuming this is true, that aspect should be emphasized and associated limitations articulated.

Answer:

The reviewer is right. The ITO patterns, to be more accurate, the gap between the ITO determines the spatial features of the deformation. This applies especially to the lateral dimension/ structure, the height of the deformation is less affected which is dominated by the amount of free volume generated in the polymer network. We have emphasized this point in the 1st paragraph on Page 3.

2. The paper alludes to haptic interface, but the changes in topography achieved are not obviously within a regime of human tactile perception. One reference in the introduction suggests that this is true, but the matter should be addressed more rigorously. The last word of the manuscript is "Braille", suggesting that the findings will be enabling for dynamic Braille, but the difference in topography amplitude between standard Braille and this paper are about 6 or 7 orders of magnitude. As such, use of the term so loosely should be avoided and some rationale for the potential provided.

Answer:

The reviewer is right that the first reference suggests that the dimension for the human tactile feedback is in the order of nanometers. Based on this study, the dimension of our current topographic patterns is sufficient for the haptic feedback. However, for the braille display as mentioned in the introduction, it might not be large enough. We are now at the stage of performing study on human perception of surface structures with various patterns/ dimensions in both static and dynamic way. The results will provide us with the guidance of designing the structures. We have changed the text to describe the braille display as our prospect and the limitations that we still have in the Discussion section on Page 14.

3. The explanation of the phenomena observed is rooted in a volumetric effect, particularly electrically (AC) induced free volume addition between interdigitated electrodes. This is plausible and backed up by molecular dynamics simulations included in supplementary matter. As no experimental evidence of a free-volume enhancement effect is presented, it would be prudent to offer language less strong about the explanation than that provided in the paper. As written, the readers may gain the false impression that the mechanism at work has been proven, which it has not.

Answer:

As the dimensions of the coating are restricted in the x-y plane, the generation of surface topographies must be based on the density reduction. The concept originates from our earlier studies on the light-induced surface topographies (ref 24) in which we have proven the increase in free volume by in situ density measurements. Therefore, in the text our tone is firm. We have now explained this better and refer to the light-driven system better on Page 3, 2nd paragraph.

4. Importantly, the effect is found to be maximized at the glass transition temperature for the LCN used, which is about 60 C. As this would not be directly useful for devices or products operating at or near room temperature, it would be helpful for the authors to indicate the

feasibility of achieving molecular designs that would yield the phenomenon at or near room temperature.

Answer:

The reviewer is right. In this work, we found out that the optimum operating temperature is 60°C, which related to the glass transition temperature. Therefore, to be more practical we need shift T_g to room temperature. This is can be achieved by a proper LC mixture which is now briefly addressed the Page, 2nd and discussed more extensively in the Discussion section on Page 14.

5. Related to (4), the authors indicate that because the observed topography switching occurs under AC (not DC) excitation that features resonance, the effect is not one of simple thermal expansion. While plausible, it would be helpful if the authors reported some proof that the application of resonant-condition AC fields did not increase the coating temperature substantially, perhaps by thermal imaging.

Answer:

This relates to the question 3 of the first reviewer. The temperature increase in the sample is measured to be 6 °C and the results are given in the Supplementary section “Sample heating and thermal effects” and the related Figure S4. We calculated based on known volumetric thermal expansion coefficients of LCNs that the corresponding thermal expansion is 6.4 nm, which is minor compared with the overall expansion of 250 nm.

Reviewers' comments:

Reviewer #1 (Remarks to the Author):

I have no further comments on the manuscript. I recommend publication of this manuscript in Nat. Commun.

Reviewer #2 (Remarks to the Author):

The authors reported the observed chaotic oscillation behavior in the surface deformation when an electric field applied. On the contrary, their numerical results do not show any chaotic behavior at all. Their numerical results only show us electric-field-induced oscillation at a frequency of the applied electric field. (Is the oscillation frequency the same as the frequency of the applied field?) They have shown us that the amplitude of oscillation is maximized when the frequency of the applied field coincides with the system resonance frequency. However, as far as I understand, harmonic oscillation at a single frequency (determined by the applied field) must be distinguished from the chaotic behavior. If the turbulent behavior in your experimental results cannot be supported by the theory or any numerical results, the observed turbulent behavior may not be a chaotic behavior but simply 'noisy' measurement results. Could you explain your understanding regarding this issue? I believe that the manuscript (including the title and abstract, if necessary) need to be revised accordingly.

Reviewer #3 (Remarks to the Author):

The authors have adequately addressed my concerns with the paper and I now recommend it for publication in Nature Communications. Some new typos have been introduced in the revision, but these can be fixed at the galley proof stage.

To address the reviewer's comments:

Remarks of the Reviewer #2:

The authors reported the observed chaotic oscillation behavior in the surface deformation when an electric field applied. On the contrary, their numerical results do not show any chaotic behavior at all. Their numerical results only show us electric-field-induced oscillation at a frequency of the applied electric field. (Is the oscillation frequency the same as the frequency of the applied field?) They have shown us that the amplitude of oscillation is maximized when the frequency of the applied field coincides with the system resonance frequency.

However, as far as I understand, harmonic oscillation at a single frequency (determined by the applied field) must be distinguished from the chaotic behavior. If the turbulent behavior in your experimental results cannot be supported by the theory or any numerical results, the observed turbulent behavior may not be a chaotic behavior but simply 'noisy' measurement results. Could you explain your understanding regarding this issue? I believe that the manuscript (including the title and abstract, if necessary) need to be revised accordingly.

Answer to the reviewer and corresponding changes in the manuscript:

The chaotic behavior is primarily an experimental observation which goes far beyond the noise level of the measurement. We indicated this more explicitly in the text of the manuscript (blue characters in page 6). We discuss, also more explicitly indicated as a possible explanation, this further on page 9 (blue characters). We removed on page 10 a second and alternative explanation based on overtones of the resonance.

As urged by the reviewer we changed the title by removing the word 'chaotic'. Similarly, we removed 'chaotic' from the abstract.

With respect to the oscillation frequency, it clearly deviates from the frequency of the AC field. This is already emphasized at the bottom of page 6/ top of page 7.

REVIEWERS' COMMENTS:

Reviewer #2 (Remarks to the Author):

The revised manuscript is acceptable in its current form.

Revision cover letter

All three referees did not raise further questions or had remarks for further improvement. Therefore, we keep the manuscript in its current form with respect to the scientific contents.